# Supporting Perinatal Mental Health and Wellbeing during COVID-19

**DOI:** 10.3390/ijerph19031777

**Published:** 2022-02-04

**Authors:** Laura Bridle, Laura Walton, Tessa van der Vord, Olawunmi Adebayo, Suzy Hall, Emma Finlayson, Abigail Easter, Sergio A. Silverio

**Affiliations:** 1Women’s Services, St. Thomas’ Hospital, Guy’s and St. Thomas’ NHS Foundation Trust, London SE1 7EH, UK; Emma.Finlayson@gstt.nhs.uk; 2Women’s and Children’s Division, Princess Royal University Hospital, King’s College Hospital NHS Foundation Trust, London BR6 8ND, UK; L.Walton1@nhs.net; 3Women’s Health Services, Chelsea and Westminster Hospital, Chelsea and Westminster NHS Foundation Trust, London SW10 9NH, UK; t.vandervord@nhs.net; 4Maternity Services, Watford General Hospital, West Hertfordshire Hospitals NHS Trust, Watford WD18 0HB, UK; Olawunmi.Adebayo@nhs.net; 5Maternity Services, University Hospital Lewisham, Lewisham and Greenwich NHS Trust, London SE13 6LH, UK; Suzy.Hall@nhs.net; 6Department of Women & Children’s Health, School of Life Course & Population Sciences, King’s College London, London WC2R 2LS, UK; Abigail.Easter@kcl.ac.uk (A.E.); Sergio.Silverio@kcl.ac.uk (S.A.S.)

**Keywords:** COVID-19, perinatal mental health, midwives, women’s health, pregnancy, childbirth, postpartum

## Abstract

Mental health is especially important as women transition into parenthood. The COVID-19 pandemic has necessitated the rapid reconfiguration of maternity services, including perinatal mental healthcare, as offered by Specialist Perinatal Mental Health Midwives, in NHS Trusts in the United Kingdom. This article represents work undertaken in rapid response to the COVID-19 pandemic and aims to document the findings from March 2020 up until May 2021 in literature published on perinatal mental health through the pandemic, as well as to engage in a knowledge mapping exercise across five NHS Trusts in London. In this research, we utilised a critical review methodology which purposefully selects and synthesises materials after extensive literature searching to provide a broad and informed narrative around an issue. For our knowledge mapping exercise, we utilised an inclusive stance to gather, pool, and synthesise data from five NHS Trusts regarding the provisions and reconfigurations of their perinatal mental health services, creating a comparable and translatable snapshot in time. Our rapid, critical review highlighted two themes: ‘Increased Perinatal Distress’ and ‘Inaccessible Services and Support’. Our knowledge mapping exercise produced four themes: ‘Retention of Existing Service Provision’; ‘Additional Services Provided’; ‘Reconfiguration of Service Provision’; ‘Additional Provision to Support Staff Wellbeing’. We conclude by offering best practice guidance in order to provide shared learning to aid the transition through para-pandemic circumstances to service delivery in a post-pandemic ‘new normal’.

## 1. Introduction

Mental ill health can affect up to 20% of women at any time between pregnancy and 12 months post birth [1], which is referred to as the perinatal period. This ranges from mild to severe and can include a variety of illness, such as depression, bipolar disorder, anxiety disorders, and postpartum psychosis [2]. Perinatal mental illness is more common in women from Minority Ethnic groups [3], and a study conducted in South London found a prevalence of mental ill health of up to 27% in women pre-COVID-19 [4]. Since these statistics represent those women who receive formal diagnoses, it is likely that the prevalence of mental ill health in the perinatal population is much higher, with many women going undiagnosed and without treatment or support. Mental ill health in the perinatal period not only has devastating effects on a woman at the time they transition to becoming a mother, but also affects those around them, including their children, partners, family, and friends, whilst also posing a considerable financial burden to health and social care services [5]. The area of perinatal mental health has become a major focus of interest in recent times, with ongoing investment in new specialist mental health services [5].

The SARS-CoV-2 (coronavirus) or COVID-19 pandemic has brought about an unprecedented threat to the health of the population, as well as a significant change to daily life. Governments across the world have imposed stay-at-home orders (or ‘lockdowns’) and introduced physical distancing and quarantine measures in an attempt to reduce the spread of the virus and ease the burden on hospitals and healthcare systems, which have experienced and continue to experience record high numbers of patients requiring in-patient intensive care [6,7]. Whilst maternity services in the United Kingdom have continued to function, the capacity at which they are delivering care has been significantly reduced, with the almost total cessation of continuity of care and home-birthing, reduced provision of face-to-face antenatal and postnatal care, and limits on birth partners during labour [8]. In the early stages of the pandemic, the Royal College of Obstetricians and Gynaecologists recommended that pregnant women should ‘shield’—that is, remain at home and under no circumstances leave unless attending a maternity care appointment, seeking urgent medical care, or attending hospital to give birth. This was later revoked, though lately there has been renewed concern that women in their third trimester may experience worse outcomes if they contract COVID-19 [9]

With the worry people have experienced about becoming ill and possibly dying from COVID-19 as well as the restrictions placed on human life, it is no wonder that “*fear from the virus is spreading even faster than the virus itself*“ [10]; p. 129). For perinatal women, the psychological distress associated with COVID-19 is expected to pose “*unprecedented challenges that can significantly impact on women’s mental health*” [11]; p. 310).

In 2018, the Royal College of Midwives (RCM) [12] released a document entitled “SPECIALIST MENTAL HEALTH MIDWIVES: What they do and why they matter”. This document highlights the role perinatal mental health midwives play in supporting mental health in pregnancy, not just for women and families, but also for the whole multidisciplinary team. Midwives hold the unique position of sharing information on an infant’s health right from the beginning of a pregnancy with the families they care for. This is an area which is sometimes forgotten in maternity, and it is crucial midwives start the conversation as early as possible.

Prior to the pandemic, perinatal mental health midwives based in London started a Pan-London Perinatal Mental Health Midwives Forum (L.B., L.W., T.v.d.V., O.A., and S.H.), supported by NHS England. Meetings were held quarterly to compare how roles were utilised, and members of the forum also acted to support one another. This helped to map out roles in terms of what each midwife was doing and whether there was uniformity in how they all worked—i.e., highlighting which Trusts were doing certain things, noting any inconsistencies and similarities, and establishing a functional network to link perinatal mental health midwives across London with those in bordering counties. This group has now grown to over 129 midwives throughout the UK and Ireland, offering a wonderful network of support during these unprecedented times. We hope that by sharing some of the ways we have changed the way we have supported women, we can help you and your service think of alternative ways to support women and families during COVID-19 and the subsequent years, which will undoubtedly be affected by this pandemic [10].

Whilst many women will experience perinatal mental health issues, the increased pressure of COVID-19 on both the public and healthcare services has meant that, understandably, maternity services have seen an increase in the incidence of poor mental health. A previous call to action [13] asserted that now is the time to prioritise women’s mental health in the public health and mental health communities.

Our aims in this study were therefore twofold: (1) To summarise the published perinatal mental health research conducted in the UK since the beginning of the COVID-19 pandemic and (2) to report on five NHS Trusts’ responses to delivering perinatal mental health care over the past year.

## 2. Materials and Methods

The work conducted as part of this article was carried out using a twofold methodology: a rapid response critical review of the perinatal mental health literature published about COVID-19 and a knowledge mapping exercise documenting the perinatal mental health service provision and reconfiguration across five NHS Trusts during the COVID-19 pandemic.

### 2.1. Rapid Critical Review of the Literature

The first part of this work presents a rapid response critical review of the literature published on perinatal mental health and COVID-19 early on during the pandemic. Critical reviews are used purposefully to synthesise materials after extensive, but not systematic, literature searching. In doing so, critical reviews provide a broad and informed narrative around an issue worthy of discussion by drawing upon specific literature for their specific contribution to the field of study (Grant and Booth, 2009). The critical review does therefore not assess for or evaluate quality, deeming all literature to contribute to the scoped knowledge base, regardless of its credibility or merit. A critical review, therefore, “*typically manifest in a hypothesis or a model, not an answer*” [14]; p. 93), more often rendering the outcome littered with more questions or new ways of thinking than when the review was first started. Within the context of this article, we present literature regarding the provision of perinatal mental health care during the COVID-19 pandemic and the effect of the pandemic on perinatal mental health. As a group, we searched individually for articles using literature databases and social media to gather publications related to the impact of COVID-19 on pregnant women and people from March 2020 up until May 2021. We searched for articles related to pregnancy, birth, the postnatal period, and COVID-19 from all over the world, with a focus on evidence arising from the UK. We did not discriminate on the basis of country of origin, type of article (e.g., empirical, review, theoretical), where it had been published, or supposed quality—all in-keeping with the guidelines for critical reviews, as explained above. We extracted any findings related to the impact of restrictions and the virus itself on the maternal mental wellbeing and service access satisfaction. We then met to review the papers as a group, inductively and iteratively sorting the ‘headline information’ from each paper into broad categories, which were then broadly synthesised before being reduced into two themes.

### 2.2. Local Knowledge Mapping Exercise

The second part of this work presents a knowledge mapping exercise, documenting responses to the pandemic (regarding perinatal mental health provision by midwives) by five NHS Trusts in London or counties bordering London in the United Kingdom. This knowledge mapping exercise draws on Trust-level policy regarding service provision and reconfiguration during the pandemic. The process of knowledge mapping has been described as proffering a mode of associating information from numerous sources with the aim of understanding the “*dynamics at play in a health system or area of work*” ([15]; p. 636). Knowledge mapping exercises are underpinned by the key principles of being “*inclusive to ensure identification of all relevant evidence*” ([16]; p. 3) whilst enabling a comparable and translatable snapshot of data [15]. The process can be broken down into five stages of data handling (as seen in Figure 1). It began with ‘Data Acquisition’, where we reported on the pandemic-related service reconfiguration by perinatal mental health midwives based in five Trusts in and around London, the United Kingdom. The second stage involved ‘Data Refinement’, where distinct thematic areas to include were agreed upon. These were: ‘Retention of Existing Service Provision’; ‘Reconfiguration of Service Provision’; ‘Additional Provision to Support Staff’. After this, we engaged with the process of ‘Data Pooling’, whereby data sourced from all five NHS Trusts were recorded in their original format. Next came ‘Data Comparison’, where data were further refined to allow for uniform reporting, after which comparisons across the dataset could be made with explainers presented, as necessary, to illustrate Trust to Trust differences. The final process entailed ‘Data Visualisation’, where the uniform data and comparisons allowed for a final version of the knowledge mapping exercise to be presented in a tabular format. By doing this, we are able to descriptively report NHS Trust-level responses for the provision of perinatal mental health care by midwives between March 2020 and May 2021—the first, second, and third waves of the COVID-19 pandemic in the UK. Whilst we have made every effort to ensure the accuracy of the information mapped at the time of writing this article, the changing nature of the pandemic means that local, national, and international guidance is changing rapidly; therefore, our knowledge mapping exercise provides a snapshot in time rather than a dynamic and all-encompassing report.

## 3. Results

The results of this work are reported in two halves. The first is dedicated to the rapid response critical review of the literature surrounding perinatal mental health care and care provision by perinatal mental health midwives during the COVID-19 pandemic, while the second reports on the findings from a knowledge mapping exercise of the provision and reconfiguration of perinatal mental health care services in five NHS Trusts during the COVID-19 pandemic.

### 3.1. Rapid Response to COVID-19 Pandemic Critical Review of the Literature

The following section describes the results of the rapid critical review, discussed as two themes: ‘Increased Perinatal Distress’ and ‘Inaccessible Services and Support’.

#### 3.1.1. Increased Perinatal Distress

Used as an umbrella term, perinatal distress describes symptoms of “*depression, anxiety and stress which women may experience from conception to one year following birth*”. The genesis of perinatal distress can be varied and multifactorial, including physical, psychological, and socio-cultural factors. Women at particular risk may have a history of mental ill health; a history of abuse or domestic violence; or may have a limited support network, which includes financial or other major life stresses [17]

The impact of the virus on pregnant women has caused an understandable intensity of distress. The initial lack of information surrounding the virus, the restrictions to social interactions, increased loneliness, women’s concerns over their own health and fear of infection and transmission of the virus to their unborn child, on top of the sudden alterations in maternity care, all will have resulted in this [18]. Perinatal distress has been seen across the globe during the pandemic in high-, middle- and low-income countries alike ([19,20,21]). Tommy’s charity conducted a survey of 1000 new or expectant mothers in the UK during the pandemic [22]. The charity found that 70% of women felt overwhelmed at some point in their pregnancy, with 14% saying they struggled throughout, 49% feeling nervous or anxious, and 77% saying the pandemic had added to their fears.

A similar story was heard from the Babies in Lockdown report published by Best Beginnings [23]. A total of 87% of parents reported an increase in anxiety during the pandemic, and one of the compounding factors was the false information and advice given to pregnant women. The fear of infection, financial insecurity, social isolation, and the restrictions put on the partners and families of women/birthing people during antenatal and intrapartum care all added to reports of anxiety, confusion, and loss. The report found that 32% of families wanted help with their mental wellbeing. Those who had the highest levels of anxiety in relation to the pandemic were Black/Black British (46%), Asian/Asian British (50%), parents under 25 (54%), and parents with household incomes of less than £16k (55%).

Similar findings have been reported worldwide through a systematic review looking at the impact of COVID-19 on mental health in pregnancy from Canada, China, Italy, Turkey and Greece [24]. The review stated that there was a statistically significant increase in levels of anxiety compared to those of depression. A thematic analysis completed in Australia looking at a popular online forum for new and expecting parents found that this heightened level of distress was significant [20]. Women were worried about attending scans or going food shopping, with one woman stating, “I feel the most anxious, overwhelmed, isolated, out of control as I’ve ever felt before.”

From pregnancy to two years postpartum is a crucial period of rapid development that sets the foundations for later health, happiness, and wellbeing [25]. It is a stage of exceptional vulnerability, when babies are especially reliant on their caregivers and very sensitive to their environment. Feedback from parents from the Babies in Lockdown report [23] stated that 25% of parents reported concerns about their relationship with their baby, and 35% of these parents felt they would like to obtain help with this. In the recent report Working for Babies [25], professionals working with families were asked about the impact of the pandemic on mental health, with 98% saying that babies had been impacted by parental anxiety/stress/depression, leading to an effect on bonding and responsive care.

#### 3.1.2. Inaccessible Services and Support

All of the reports relating to perinatal distress referenced the inability to access services or support. An international cross-sectional survey of 6894 participants in 64 different countries was conducted between 26 May 2020 and 13 June 2020 [26]. The most frequently described anxieties were related to pregnancy and birth, including family being unable to visit after birth (59%), the lack of a support person during labour (55%), and COVID-19 causing changes to their birth plan (41%). Further concerns were related to not having access to medical appointments, which is associated with a significantly higher chance of posttraumatic stress and anxiety. This was recognised pre-pandemic through a systematic review and meta-synthesis for perinatal mental health services citing such barriers as lack of resources and the fragmentation of services [27], which now faced further restrictions due to staff self-isolating and a lack of face-to-face services.

A survey conducted in the UK found that almost half of all patients seen by a specialist mental health midwife reported their support to have stopped due to pandemic [28]. This was more frequently seen by first-time mothers, with one reporting, ‘*It’s hard to feel like you’re important, really just feel like we’re inconveniencing the hospital’* ([28]; p. 13). Further discussion showed the impact when these services were not available or taken away, “‘*devastated. Frightened. Powerless. Helpless. Shocked. I was having an elective c-section due to previous birth trauma. To have my support taken away 2 days before I went into theatre and be told I had to do it alone feels like it is the hardest thing I have ever been asked to do*.’” ([28]; p. 3).

The Best Beginnings [23] survey of women in the UK highlighted a lack of face-to-face appointments. “My anxiety is through the roof and I am trying to get professional help with it to manage, but I’ve been told there is a long waiting list.” Only 32% of those surveyed were confident that they could find mental health support if they needed it.

Perinatal mental health teams faced similar challenges, with the majority of service delivery needing to be delivered online ([29,30]); however, the importance of face-to-face assessment in high-risk circumstances was emphasised by the rapid report into maternal deaths in the UK during the first three months of the pandemic, which included four suicides and two domestic homicides [31].

A study featured in the British Medical Journal [32] found that there was a notable increase in some Trusts restricting access for women requesting elective caesarean sections (ELCS) for non-medical reasons during the COVID-19 pandemic. This is concerning when considering the experiences many women have endured which have led them to this decision. Most women with secondary tokophobia requesting an ELCS will have a history of trauma during birth, and those women with primary tokophobia could have suffered a history of sexual trauma [33]. The potential impact on a woman’s mental health if she is denied the opportunity to birth how she chooses is significant and the denial of this choice could not only cause further trauma but may also have long-term impacts on a woman’s mental health.

In a survey by Tommy’s [22], the majority of women reported hearing false information and misconceptions regarding pregnancy and birth. It is also important to consider that many women have requested ELCS during the pandemic as a result of fear of their partner not being at the birth due to visiting restrictions and increased anxiety surrounding childbirth. With virtual appointments being put in place in order to reduce the risk of pregnant women being exposed to COVID-19 during antenatal appointments, it could be said that women have had fewer opportunities to discuss their fears regarding birth, whilst the reduced continuity of carer could have impacted on women’s ability to feel comfortable with being honest about their anxieties.

Lockdown measures have also meant that women and birthing people needed to remain inside for long periods of time, and this sadly saw an increase in domestic violence [21]. Midwives who would usually be able to see women face to face from as early as 6 weeks were now not seeing women in clinics until 28 weeks into their pregnancy. This reduced their ability to safely enquire about domestic violence and offer support in a timely manner. The rise in domestic violence calls for support from police to helplines rose worldwide [34]).

Working women have faced unfair treatment in the workplace during the pandemic. Pregnant Then Screwed [35] conducted an online survey of 19,950 mothers and pregnant women from 16 to 18 July 2020. The data showed that 15% of mothers either had been made redundant or expected to be made redundant and, of those, 46% said that a lack of childcare provision played a role in their redundancy. Worse still, the survey found that 45% of pregnant workers who were working outside of the home had not had an individual risk assessment conducted, rising to 52% for Black, Asian, and Ethnically Diverse pregnant women, who we know are at increased risk of severe COVID-19 illness [36]. All of this has impacted their ability to access services, with women not only being concerned about losing their job, but also being at greater risk of infection if they were having to travel using public transport and then being made to sit in waiting rooms before their scheduled antenatal appointments.

### 3.2. Knowledge Mapping Exercise

The section which follows reports the results of the local knowledge mapping exercise, detailing the perinatal mental health provision by Specialist Midwives at five NHS Trusts in response to COVID-19. Findings are presented under four distinct themes: ‘Retention of Existing Service Provision’; ‘Additional Services Provided’; ‘Reconfiguration of Service Provision’; ‘Additional Provision to Support Staff Wellbeing’. These are explained simplistically in Table 1 and narratively in the prose below.

#### 3.2.1. Retention of Existing Service Provision

Retention of existing services extended to the assurance of face-to-face appointments being offered to all from the beginning of pregnancy to discharge from maternity care, if that is what women wanted. Further, perinatal mental health midwives continued to work alongside caseloading midwives, where caseloading models of care were retained. It should be noted, however, that every Trust involved in this paper saw caseload/continuity of carer teams disbanded and/or used to cover areas with poor staffing levels. Caseload teams differ to traditional midwifery care in that they offer a one-to-one continuity of care throughout pregnancy and birth.

#### 3.2.2. Additional Services Provided

During the pandemic, it was noticed there was a marked increase in the need for ad hoc support for women suffering with anxiety at all Trusts. At Trust A, this rose by 62%, while it rose by 40% at Trust C. All midwives contributing to this knowledge mapping exercise increased their availability either through increased hours, staffing, or ad hoc support.

There was also a noted increase in requests for elective caesarean sections (ELCS). Trust B responded to this surge through the specialist midwife for mental health offering a ‘Birth Without Fear’ class. This was intended to enable women whose anxiety was influencing their decisions around birth to have access to specialist mental health support and evidence-based information, ensuring they were able to make an informed choice about their birth.

The Pan-London Perinatal Mental Health Midwifery Forum developed a resource we could share with women and staff that gave information about COVID-19, seeking mental ill health support, resources to share with children, online support groups, and online exercise classes for pregnant women and staff.

#### 3.2.3. Reconfiguration of Service Provision

All specialist perinatal mental health midwives involved in this paper adapted their services to allow for further ad hoc support as needed with the increased levels of anxiety. This included adapting existing provision to either partial or fully online provision to counter any concerns about exposure to infection from having to come into the hospital. This not only led to an anecdotal reduction in levels of anxiety but also enabled patients to inquire safely about domestic violence and signposted them to services early on.

#### 3.2.4. Additional Provision to Support Staff Wellbeing

Each Trust was supportive in some way, with in-house support aimed at maintaining and improving staff wellbeing. This varied from relaxation rooms to links for online support and to more specialist services. Trust E went one step further. The local Parent–Infant Psychotherapy (PIP) service together with a perinatal psychologist also worked with the perinatal mental health midwife to set up a service to support all maternity staff at the height of the first wave of the pandemic. The project was aimed at giving maternity staff the opportunity to contact a member of the PIP team by e-mail or phone if they wanted to seek advice or support following their shift if they felt they had particularly challenging cases. This was not a referral service for patients, but more a service to give staff the opportunity to reflect and debrief with a trained professional.

The Pan-London Perinatal Mental Health Midwives Forum also adapted in light of what was happening with COVID-19. Previously meeting every three months and face to face, we changed this to be a virtual monthly drop-in. This opportunity to check in on one another and share what each service was doing was hugely influential in the changes we could then implement within our own Trusts.

## 4. Discussion

### 4.1. Summary of Main Findings

The COVID-19 pandemic has caused an unprecedented health system shock which has required the significant reconfiguration of the health service in order to continue to provide healthcare whilst also reducing the risk of viral transmission. For maternity care, guidance has been produced consistently by the RCOG with regard to how best to provide maternity care and updating advice on the risk posed by the virus to pregnant and postpartum women [37]. Maternity services have therefore been subject to repeated service-level reconfiguration [8], rendering many women unhappy with the care they received [38]. Specifically, it has been reported that women (in the general population) in the UK have experienced some of the highest levels of perinatal mental ill health ever recorded empirically [39,40] and were adversely affected by restrictions associated with COVID-19 [41,42], especially with regard to missing out on social and healthcare professional support [43].

Women and birthing people have faced unparalleled change at a time in their life which should usually allow for preparation for parenthood. This preparatory work in the lead up to the transition into parenthood should normally be accomplished with the support of healthcare professionals, including a midwife and health visitor. For women with mental ill health, this should be further supported by a specialist perinatal mental health midwife [12]. Despite numerous publications [37,44,45] recommending the continuity of care(r) and services, COVID-19 disrupted this support for many families, who were left to navigate their maternity care through a screen on their laptop and/or birth alone amongst strangers. This mapping exercise showed that the work the perinatal mental health midwives were able to adapt allowed for care to continue in as normal a way as possible, thus combatting the issues arising in perinatal mental health due to COVID-19 (which were identified in the critical review). This was only due to ongoing communication and support from management, the personal and individual drive from the perinatal mental health midwives themselves, and the peer support between members of the Pan-London Perinatal Mental Health Midwife Forum. Future and ongoing planning needs to include the ability to co-produce resources, to communicate with women and families, and for specialist midwives to remain in place to ensure that perinatal mental health care is delivered efficaciously and with the compassion it requires.

### 4.2. Strengths, Limitations, and Future Directions

To our knowledge, this is the first paper to document how services in the United Kingdom (NHS Trusts) adapted their perinatal mental health care provision, as offered by specialist perinatal mental midwives, for women and birthing people. However, we realise that this knowledge mapping exercise was limited to five predominantly London-based NHS Trusts and therefore may not be broadly or directly applicable to perinatal mental health services in other regions or countries. Another limitation linked to this one is the fact we that were unable to provide individual Trust-level statistics due to the potential for identifiability. This is the nature of local projects, and future research could rectify this using national survey methodologies, as have previously been used [8]. Future work should include a wider spectrum of Trusts throughout the UK and a wider global mapping of perinatal mental health services to see where services continued and/or adaptations were successful and ultimately share learning and best practices.

Combined with this knowledge mapping, we also offer a rapid critical review of the literature published in the early stages of the pandemic and broadly synthesise the landscape of perinatal mental health and how it was affected by the onset of COVID-19. Whilst we provide a rapid review of the work published, the pandemic has led to many empirical explorations of maternal and child health, care, and service provision. Consequently, further systematic reviews with specific questions are required to provide a more thorough insight into the effects of the pandemic on perinatal mental health services and care provision.

### 4.3. Conclusions

The impact of the COVID-19 pandemic is still not fully understood, but, as seen within services in London and based on articles from papers published early in the pandemic, increased levels of perinatal distress and restrictions to perinatal mental health services have been witnessed globally. Ongoing and global evaluations of service configurations similar to ours will be helpful as we move from the pandemic into post-pandemic recovery, building better services in what has been dubbed the “new normal”. In doing so, we attempt to do our utmost to protect our future populations through shared learning, best practice, and adaptations in the way we provide care.

## Figures and Tables

**Figure 1 ijerph-19-01777-f001:**
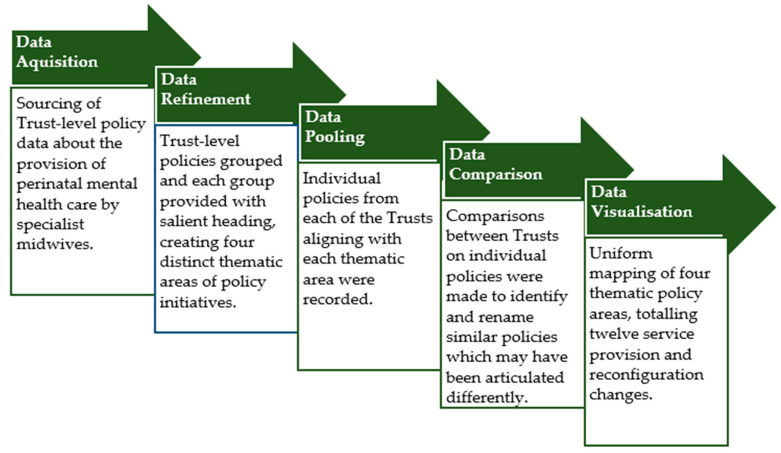
Knowledge mapping exercise process (adapted from Ebener et al., 2006) [15].

**Table 1 ijerph-19-01777-t001:** Reconfiguration of perinatal mental health service provision by trust.

Theme	Service Provisions and Reconfigurations	Trust A	Trust B	Trust C	Trust D	Trust E
Retention of Existing Service Provision	Continuation of face-to-face appointments for women with a current diagnosis of mental illness	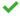	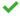	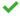	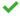	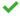
Perinatal Mental Health Midwives working alongside Caseloading Midwives	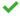	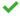	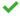	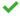	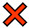
Additional Services Provided	Additional support provided (to meet increased reporting of perinatal anxiety)					
Provision of co-developed resource offering virtual support for women and/or families					
Provision of ‘Birth Without Fear’ class to meet rise in elective caesarean section requests	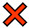		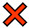	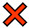	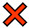
Increased Perinatal Mental Health Midwife staffing to meet demand on service	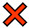	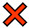	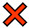	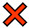	
Reconfiguration of Service Provision	Face-to-face exercise classes moved to on-line provision	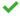	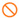	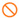	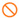	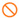
Art psychotherapy moved to online provision	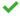	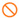	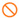	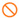	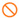
Virtual antenatal clinics offering continuity of (midwife) carer	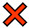		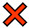	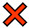	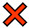
Provision of support for partner/parent to accompany woman/birthing person in exceptional circumstances only	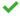	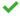	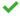	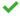	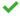
Provision of virtual appointments for those unable or not wanting to attend face-to-face					
Additional Provision to Support Staff Wellbeing	In-house support for staff mental wellbeing					

**Key:**
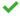
 = yes: continuation of service provision; 

 = yes: addition of service provision; 
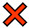
 = no; 
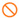
 = not applicable.

## Data Availability

Not applicable.

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
