# Peer review of "Supporting Perinatal Mental Health and Wellbeing during COVID-19"

_ijerph, 2022, doi:10.3390/ijerph19031777_

Round 1

Reviewer 1 Report

Thank you for the opportunity to review the manuscript entitled: Supporting perinatal mental health and wellbeing during COVID-19. This manuscript presents a rapid critical review of literature describing the impacts of COVID-19 on perinatal mental health and service provision. It also incorporates a mapping exercise of service provision across 5 NHS Trusts, primarily in London UK. Given the recognised importance of maternal mental health throughout the perinatal period, and the implications associated with COVID-19, this manuscript provides a valuable and useful contribution to knowledge.

The strengths of the manuscript are that the authors have compiled literature describing the impact of COVID-19 on perinatal mental health, and provided a more  localised examination of actual service provision, change, and addition in response to the pandemic. I have only very minor queries and suggested alterations/additions to propose.

Introduction

  • Page 2, beginning line 43: When outlining the prevalence of mental health difficulties, it might be helpful to also describe associated impacts. Similarly, it would also be useful to insert a sentence outlining that such prevalence only encompasses those with diagnosed mental health disorders. There will be many more with subthreshold symptoms, experiencing significant difficulty, which emphasises even more the importance of understanding impacts and methods of mitigation
  • Very minor spelling mistake on p2 line 64 (‘care’ not ‘are’)
  • In the Background and Context section (2.1), could the authors be more specific about ‘further afield’ (p2 line 89)
  • Section 2.2 (Rationale and Aim) begins with a sentence that could be streamlined/improved for clarity. I am not certain that ‘states’ is the correct word to use here

Method

  • The rationale for the use of critical review is clear (Methods section 2.1). Regardless, specificity in how this review was conducted is required. How was social media utilised? What was deemed to be ‘literature’, any inclusion/exclusion criteria? Any restriction on geographical location? I would presume not as the results present literature on an international basis, however some further, brief, details here about the procedure undertaken would be beneficial
  • Similarly, on Page 3 line 131– how were these themes agreed upon?
  • In the description of the methodology used for the knowledge mapping exercise, it would be very helpful to have some additional context/description of the Trusts involved (e.g., annual birth rate). Specific location of the Trust may not be appropriate, however any indications of region in the UK would be useful to aid interpretation of findings as currently this is described as ‘predominantly’ London Based
  • It is noted that the time period under review for the knowledge mapping exercise was March 2020- May 2021. Some Trusts saw varying provision throughout this time. Could the authors clarify whether the provision as recorded was provided uniformly across this timepoint for each Trust, or was there variation and, if so, to what extent did this vary?

Results

  • The findings from the rapid review read very well and appear comprehensive. I have just one suggestion - page 5 line 218, this is a striking statistic and would benefit from some further description (e.g., nature of impact? How recorded?)
  • Some minor re-wording on page 6 lines 266 would be useful (i.e., fewer opportunities)
  • Page 6 lines 284-285 refers to ‘sitting in waiting rooms’. This appears a little disjointed from the overall paragraph, given the focus on work lives. If this relates to a source from the review, perhaps a citation here is required, in addition to some further explanation for referring to waiting rooms.
  • Section 3.2.2 provides some interesting percentages for the increase in requirement for ad hoc support. Would it be feasible for the authors to provide this information also for the second paragraph here (page 8, line 311) for increased ELCS provision? This is pertinent given that this increase is described in the manuscript as a ‘surge’.
  • Could the authors insert information about the nature of the resources described (e.g., the Birth Without Fear resource, and those provided by the Pan-London Perinatal Mental Health Midwifery forum). Were these online resources, or in-Trust sessions? It may also be beyond the scope of the current manuscript, however any indications of uptake to these resources would be insightful

Discussion

  • It would be useful to see the authors discuss the findings as highlighted in the critical review, in light of the findings of the mapping exercise. This is referred to (although check the sentence clarity of page 9 line 373 beginning ‘The work the perinatal mental health midwives were…”), but some further consideration here would help to bring the two sections of the manuscript together
  • Page 10 line 407 – “a” limb-out?
  • Could the authors double-check the final subheading of the manuscript, as this appears to repeat the section before (strengths, limitations and future directions)

Overall I would like to congratulate the authors on this interesting and valuable insight into the impact of COVID-19 on perinatal mental health and service provision.

Author Response

Reviewer 1’s comments

1

Thank you for the opportunity to review the manuscript entitled: Supporting perinatal mental health and wellbeing during COVID-19. This manuscript presents a rapid critical review of literature describing the impacts of COVID-19 on perinatal mental health and service provision. It also incorporates a mapping exercise of service provision across 5 NHS Trusts, primarily in London UK. Given the recognised importance of maternal mental health throughout the perinatal period, and the implications associated with COVID-19, this manuscript provides a valuable and useful contribution to knowledge.

We would like to thank the reviewer for such a positive appraisal of our manuscript and for the time they have taken to recommend edits and revisions. We believe these changes have helped to improve the paper.

2

The strengths of the manuscript are that the authors have compiled literature describing the impact of COVID-19 on perinatal mental health, and provided a more localised examination of actual service provision, change, and addition in response to the pandemic. I have only very minor queries and suggested alterations/additions to propose.

We are very appreciative of the reviewer’s suggestions and have worked hard to ensure our manuscript has been revised in-line with both reviewers’ comments.

3

Introduction: Page 2, beginning line 43: When outlining the prevalence of mental health difficulties, it might be helpful to also describe associated impacts. Similarly, it would also be useful to insert a sentence outlining that such prevalence only encompasses those with diagnosed mental health disorders. There will be many more with subthreshold symptoms, experiencing significant difficulty, which emphasises even more the importance of understanding impacts and methods of mitigation

Thankyou for this suggestion. We agree this will be useful information to included a sentence to this effect and have also added a sentence on burden, as below:

“Whilst these statistics represent those women who receive formal diagnoses, it is likely the prevalence of mental ill health in the perinatal population is much higher, but many women go undiagnosed, and without treatment or support. Mental ill health in the perinatal period not only has devastating effects to a woman at the time they transition to becoming a mother, but also affects those around them including their children and partners, family and friends, whilst also posing a considerable financial burden to health and social care services (5). The area of perinatal mental health has become a major focus of interest in recent times, with investment in new specialist mental health services (5).”

4

Introduction: Very minor spelling mistake on p2 line 64 (‘care’ not ‘are’)

Thankyou for pointing this out. Now corrected.

5

Introduction: In the Background and Context section (2.1), could the authors be more specific about ‘further afield’ (p2 line 89)

Thankyou for highlighting this. We have changed ‘further afield’, to ‘bordering counties’ for specificity and clarity, without being directly identifiable.

6

Introduction: Section 2.2 (Rationale and Aim) begins with a sentence that could be streamlined/improved for clarity. I am not certain that ‘states’ is the correct word to use here

Apologies for this. The sentence was rendered confusing by the change in citation style. We have amended the sentences to read as follows:

Whilst many women will experience perinatal mental health issues, the increased pressure of COVID-19 on both the public and healthcare services has meant that understandably maternity services, have seen an increase in poor mental health. A previous call to action (13) has asserted that now is the time to prioritise women’s mental health in the public health and mental health communities.

7

Method: The rationale for the use of critical review is clear (Methods section 2.1). Regardless, specificity in how this review was conducted is required. How was social media utilised? What was deemed to be ‘literature’, any inclusion/exclusion criteria? Any restriction on geographical location? I would presume not as the results present literature on an international basis, however some further, brief, details here about the procedure undertaken would be beneficial

Thankyou for this point. We have added in a clarification to allow further understanding, as follows:

We searched for articles related to pregnancy, birth, postnatal, and COVID-19 from all over the world, with a focus on evidence arising from the UK. We did not discriminate on the basis of country of origin, type of article (e.g., empirical, review, theoretical), where it had been published, or supposed quality – all in-keeping with the guidelines for critical reviews, as explained above.

8

Method: Similarly, on Page 3 line 131– how were these themes agreed upon?

Thanks for this, we have clarified as follows:

We then met to review the papers as a group, inductively and iteratively sorting the ‘headline information’ from each paper into broad categories, which were then broadly synthesized before being reduced into two themes.

9

Method: In the description of the methodology used for the knowledge mapping exercise, it would be very helpful to have some additional context/description of the Trusts involved (e.g., annual birth rate). Specific location of the Trust may not be appropriate, however any indications of region in the UK would be useful to aid interpretation of findings as currently this is described as ‘predominantly’ London Based

We would like to thank the reviewer for this comment. We have added some clarity by stating that the Trusts were either in London or in counties bordering London, but for reasons of Trust-level information and identifiability we are not permitted to provide any such information, which we agree is a shame. We hope you understand our predicament, and we will explain this as a limitation in lieu of this issue.

10

Method: It is noted that the time period under review for the knowledge mapping exercise was March 2020- May 2021. Some Trusts saw varying provision throughout this time. Could the authors clarify whether the provision as recorded was provided uniformly across this timepoint for each Trust, or was there variation and, if so, to what extent did this vary?

This information should be taken to mean the uniform application, with any variation otherwise explained throughout the manuscript. We chose to focus on this period, as it encompassed the worst of the COVID-19 waves/peaks/variants we have seen to date and also before the real easing of lockdown restrictions.

11

Results: The findings from the rapid review read very well and appear comprehensive. I have just one suggestion - page 5 line 218, this is a striking statistic and would benefit from some further description (e.g., nature of impact? How recorded?)

We agree this is a striking statistic. We have amended the sentence to add some clarity, but as this study was undertaken by another team, we only report on their report and do not have the full detail of their findings. These can however be found through the associated reference.

12

Results: Some minor re-wording on page 6 lines 266 would be useful (i.e., fewer opportunities)

Thankyou for raising this. We have amended the paragraph as follows:

The majority of women in a survey by Tommy’s (22) reported hearing false information and misconceptions regarding pregnancy and birth. It must also be considered how many women have requested ELCS during the pandemic, as a result of fear of their partner not being at the birth due to visiting restrictions and increased anxiety surrounding childbirth. With virtual appointments being put in place in order to reduce the risk of pregnant women being exposed to Covid-19 during antenatal appointments, it could be said that women have had fewer opportunities to discuss their fears regarding birth, whilst reduced continuity of carer could have impacted on women’s ability to feel comfortable to be honest about their anxieties.

13

Results: Page 6 lines 284-285 refers to ‘sitting in waiting rooms’. This appears a little disjointed from the overall paragraph, given the focus on work lives. If this relates to a source from the review, perhaps a citation here is required, in addition to some further explanation for referring to waiting rooms.

Thankyou for raising this point. We have clarified the paragraph as follows:

Worse still, the survey found that 45% of pregnant workers who were working outside of the home had not had an individual risk assessment conducted, rising to 52% for Black, Asian, and Ethnically Diverse pregnant women, who we know are at increased risk of severe COVID-19 illness (37). All of which impacted on ability to access services when women were not only concerned about losing their job, but were at greater risk of infection if they were also having to travel using public transport and were then made to sit in waiting rooms before their scheduled antenatal appointments.

14

Results: Section 3.2.2 provides some interesting percentages for the increase in requirement for ad hoc support. Would it be feasible for the authors to provide this information also for the second paragraph here (page 8, line 311) for increased ELCS provision? This is pertinent given that this increase is described in the manuscript as a ‘surge’.

As previously mentioned, we are, unfortunately, not permitted to share individual Trusts’ statistics. We will be sure to explain this as a limitation in lieu of this issue.

15

Results: Could the authors insert information about the nature of the resources described (e.g., the Birth Without Fear resource, and those provided by the Pan-London Perinatal Mental Health Midwifery forum). Were these online resources, or in-Trust sessions? It may also be beyond the scope of the current manuscript, however any indications of uptake to these resources would be insightful

Thankyou for raising this point. The reviewer is correct, that detail about these resources and/or evaluation of those resources is beyond the scope of this manuscript and would also lead to potential Trust identification which we have already raised as an issue and limitation. We thank the reviewer for their understanding on this matter.

16

Discussion: It would be useful to see the authors discuss the findings as highlighted in the critical review, in light of the findings of the mapping exercise. This is referred to (although check the sentence clarity of page 9 line 373 beginning ‘The work the perinatal mental health midwives were…”), but some further consideration here would help to bring the two sections of the manuscript together

Thankyou for this suggestion. We have clarified the sentence and also added to the paragraph to draw together these two distinct parts of our exercises which form this manuscript.

17

Discussion: Page 10 line 407 – “a” limb-out?

Apologies for this – we have corrected this to “…as we climb-out…”

18

Discussion: Could the authors double-check the final subheading of the manuscript, as this appears to repeat the section before (strengths, limitations and future directions)

Apologies for this. Due to a typo, the conclusion section was mislabelled as ‘4.3 Strengths, Limitations, and Future Directions’. We have rectified this making the final paragraph our ‘Conclusions’ section.

19

Overall I would like to congratulate the authors on this interesting and valuable insight into the impact of COVID-19 on perinatal mental health and service provision.

We would once again like to thank the reviewer for being so positive about our manuscript and thank them again for their suggested revisions.

Reviewer 2 Report

Thanks for recommending me as a reviewer. This paper represents work undertaken in rapid response to the COVID-19 pandemic and aimed to document the findings from March 2020 up until May 2021 in published literature on perinatal mental health through the pandemic, and also to engage in a knowledge mapping exercise across five NHS Trusts in London. In this paper, authors utilized critical review methodology which purposefully selects and synthesizes materials after extensive literature searching, to provide a broad and informed narrative around an issue. If the authors complete the revision, the quality of the study will be further improved.

  1. The introduction section is well written. I suggest that the authors remove the subheadings from the introduction section.
  2. line 134-162: If the authors describe the "local knowledge mapping exercise" more specifically in the Methods section, it can help readers understand.
  3. line 383: "4.2 Strengths, Limitations, and Future Directions" and "4.3 Strengths, Limitations, and Future Directions" have the same subheading.
  4. Authors should add limitations to the discussion section.
  5. There is no concluding section.

Author Response

Reviewer 2’s Comments

1

Thanks for recommending me as a reviewer. This paper represents work undertaken in rapid response to the COVID-19 pandemic and aimed to document the findings from March 2020 up until May 2021 in published literature on perinatal mental health through the pandemic, and also to engage in a knowledge mapping exercise across five NHS Trusts in London. In this paper, authors utilized critical review methodology which purposefully selects and synthesizes materials after extensive literature searching, to provide a broad and informed narrative around an issue. If the authors complete the revision, the quality of the study will be further improved.

We would like to extend our thanks to the reviewer for their appraisal of our manuscript and for the suggested changes to further improve it. We believe these changes have indeed strengthened the script.

2

The introduction section is well written. I suggest that the authors remove the subheadings from the introduction section.

Thankyou for this. We have now removed the sub-headings as per your suggestion.

3

line 134-162: If the authors describe the "local knowledge mapping exercise" more specifically in the Methods section, it can help readers understand.

Thankyou for raising this. We have added a figure to assist our explanation and hope that this addresses the point made by the reviewer.

line 383: "4.2 Strengths, Limitations, and Future Directions" and "4.3 Strengths, Limitations, and Future Directions" have the same subheading.

Apologies for this. Due to a typo, the conclusion section was mislabelled as ‘4.3 Strengths, Limitations, and Future Directions’. We have rectified this making the final paragraph our ‘Conclusions’ section.

4

Authors should add limitations to the discussion section.

Thankyou for pointing this out. We have now added some limitations as follows:

However, we realise the knowledge mapping exercise is limited to five, predominantly London-based NHS Trusts and therefore may not be broadly or directly applicable to perinatal mental health services in other places, or globally. Another limitation linked to this one, is the fact we are unable to provide individual Trust-level statistics due to the potential for identifiability. This is the nature of local projects, and future research could rectify this using national survey methodologies as has previously been undertaken (8).

5

There is no concluding section.

As above, we would like to apologise for this typo, which meant the conclusion section was mislabelled as ‘4.3 Strengths, Limitations, and Future Directions’.   We have rectified this making the final paragraph our ‘Conclusions’ section.

Round 2

Reviewer 2 Report

The authors have faithfully completed the revision.